# Prevalence and Predictors of Positive Screening of Body Dysmorphic Disorder in Eastern Saudi Women Seeking Cosmetic Procedures: Implications for Clinical Practice in the Social Media Era

**DOI:** 10.3390/healthcare13243232

**Published:** 2025-12-10

**Authors:** Anfal Mohammed Alenezi, Bandar Abdulrahman Mansour AlQahtani, Ashokkumar Thirunavukkarasu, Hatim Alrashed, Boshra Abdullrahma H. Alsardi, Tamam Abdulrahman B. Aldaham, Rahmah Mohammed D. Alsabilah, Khulud Najeh N. Alazmi

**Affiliations:** 1Department of Surgery, College of Medicine, Jouf University, Sakaka 72388, Aljouf, Saudi Arabia; 2Department of Family and Community Medicine, College of Medicine, Jouf University, Sakaka 72388, Aljouf, Saudi Arabia; bandarbkq9989@gmail.com (B.A.M.A.); ashokkumar@ju.edu.sa (A.T.); 3College of Medicine, Jouf University, Sakaka 72388, Aljouf, Saudi Arabiabushra10012003@gmail.com (B.A.H.A.); t.aldaham0@gmail.com (T.A.B.A.); rahmarm7910@gmail.com (R.M.D.A.); yasalazzmi.55@gmail.com (K.N.N.A.)

**Keywords:** body dysmorphic disorders, cosmetic procedures, social media, women’s health, Saudi Arabia

## Abstract

**Background/Objectives**: Social media popularity and shifting cultural standards of beauty have intensified the growing demand for cosmetic procedures in Saudi Arabia, and body dysmorphic disorder (BDD) has not previously been given due consideration in cosmetic surgery-seeking behavior. This study determined the proportion of females screening positive for BDD and identified its predictors. **Participants and methods**: The present study employed an analytical cross-sectional design and was conducted from January 2025 to July 2025 in the Dammam region (Eastern) of Saudi Arabia. A validated tool was utilized to assess sociodemographic characteristics, cosmetic surgery history and intentions, social media exposure, and BDD symptoms. We applied a multivariate analysis to identify the predictors. **Results**: Among the 250 participants, 72 (28.8%) screened positive for BDD. The positive screening for BDD was significantly higher among the participants with a qualification of university and above (ref: up to high school, AOR = 1.291, 95% CI = 1.016–1.667, *p* = 0.038), who considered cosmetic surgery during their current visit (ref: no, AOR = 3.123, 95% CI = 1.671–4.982, *p* = 0.001), and more than 3 h use of social media (ref: less than 1 h, AOR = 4.368, 95% CI = 3.570–5.134, *p* = 0.007). **Conclusions**: The present findings suggest that a BDD screening program and a multidisciplinary approach are required to ensure ethical practice and to decrease the repeated or unnecessary interventions. Furthermore, future multicenter and mixed-method studies should be conducted to confirm these findings and guide national psychological assessment practices in aesthetic medicine.

## 1. Introduction

In recent years, the pursuit of physical beauty has gained immense prominence, driven by societal expectations, media influences, and the rise in social media platforms [1,2]. Globally, demand for both surgical and non-surgical cosmetic procedures has increased substantially as these treatments become more accessible. The rise in aesthetic procedures among the general public results from technological progress, along with minimally invasive procedures that have become more affordable [3,4,5]. Cosmetic enhancement benefits many people through self-confidence enhancement, yet specific individuals with body dysmorphic disorders (BDD) exhibit psychological disorders that drive their desire for aesthetic procedures [6,7].

BDD manifests as a psychiatric disorder that causes people to focus too intensely on imperfections of their bodies, which others mostly fail to see. The psychiatric condition of BDD resides within obsessive-compulsive and related disorders, according to the Diagnostic and Statistical Manual-5 (DSM-5) [8,9]. People who have BDD habitually inspect mirrors and spend a lot of time grooming themselves, as well as pinching their skin and persistently asking for assurance about their appearance. Among those with BDD, obsessive thinking creates substantial emotional stress that interrupts social development and occupational performance, along with emotional well-being [10,11,12,13]. People who suffer from BDD experience extreme body image distress, which constitutes a psychological disorder because the condition severely disrupts their daily functioning in life. BDD presents severe symptoms that scientists link to high levels of anxiety, as well as depression and suicidal thoughts [14,15].

Research shows that BDD affects a larger percentage of patients who undergo cosmetic surgery procedures when compared to individuals in the general public [4,16]. People with BDD show higher statistics for undergoing dermatological plastic and aesthetic surgeries because they think their perceived defects will decrease their distress levels [16]. Cosmetic procedures produce short-lived satisfaction rates among people who have BDD. After cosmetic surgery, most patients persist in noticing their flaws and feeling unsatisfied with the surgical results, which drives them to repeatedly request new procedures [17]. Preoperative psychological screening plays an essential role in identifying patients at risk because it helps direct suitable mental health treatments to BDD patients instead of exposing them to pointless surgical procedures [18,19].

Social media platforms have a strong influence on how people perceive their body image, particularly among young adults and adolescents. Users on Instagram, TikTok, and Snapchat encounter digitally enhanced presentations of beauty standards that defy reality. Social media influencers introduce cosmetic procedures to users through their posts, which creates a trend towards interventions that enhance their aesthetic outcomes, leading to increased demand [20,21]. Societies across cultural groups deeply embed their beauty ideals into social standards, creating ongoing pressure, especially for women, to match these standards. Different cultures have various preferences regarding idealized body and face appearance, yet social conventions agree that specific physical characteristics are considered most appealing [22,23].

The beauty norms in Middle Eastern nations, and Saudi Arabia in particular, are unique and reflect both native cultural values and international media exposure. More people now accept aesthetic procedures due to the accessibility of cosmetic procedures and changing social attitudes toward these treatments [24,25,26]. The worldwide increase in demand for cosmetic procedures coexists alongside a limited understanding of the psychological elements that motivate people to undergo aesthetic modification. Research focusing on BDD and cosmetic surgery is conducted in international studies, although the majority of investigations have not been conducted in Middle Eastern and Arab populations, especially in Saudi Arabia. The literature review found a significant lack of studies regarding BDD and its association with cosmetic surgery-seeking behavior among Saudi female populations. Additionally, in the continually evolving social media era, it is crucial to obtain updated information on this public health issue. Furthermore, determining the most frequently sought cosmetic surgeries and identifying predictors of BDD among those seeking treatment is critical in planning cost-effective, evidence-based, and tailored strategies. Many cosmetic surgery facilities do not routinely screen patients psychologically, making this research important for identifying gaps in current practice. Undiagnosed BDD may lead some patients to undergo unnecessary or repeated procedures. A screening system would boost patient results by giving substantial BDD symptom sufferers proper mental care in place of useless surgical treatments. The study is necessary for public health goals because it focuses on the psychological state of individuals undergoing cosmetic treatments. Therefore, the present study aimed to determine the proportion of females screening positive for BDD among those seeking cosmetic surgeries. Furthermore, we aimed to determine the predictors of BDD among them.

## 2. Materials and Methods

### 2.1. Study Description

The present study employed an analytical cross-sectional design and was conducted from March 2025 to July 2025 in the Dammam region (Eastern) of Saudi Arabia. The study setting consisted of private and government outpatient clinics specializing in dermatology, cosmetic surgery, and plastic surgery. In this study, cosmetic procedures were defined as elective aesthetic treatments sought primarily to enhance appearance, including both surgical and non-surgical interventions.

### 2.2. Inclusion and Exclusion Criteria

This research included Saudi females aged 18 years and older who sought cosmetic procedures at the selected clinics (dermatology, cosmetic, and plastic surgery) during the study period. The clinics we included were both government and private clinics. Patients presenting for either surgical or non-surgical cosmetic treatments were invited to participate. We excluded the individuals who were unable to provide informed consent, those with acute psychiatric illness, or cognitive impairment that could interfere with participation, as well as those seeking procedures primarily for functional, reconstructive, or medical reasons rather than cosmetic purposes.

### 2.3. Sampling Procedures

We have used an online sample size calculator to determine the minimum required sample size for females seeking cosmetic surgery [27]. In the online sample size calculator, we set the confidence interval to 95%, the margin of error to 5%, and the population size to infinite. The expected proportion was set as 19.2% based on previous literature [28]. After entering these values in the online calculator, we found that 239 participants were required to achieve a valid conclusion. The study employed a non-probability convenience sampling method to recruit participants. Participants were recruited consecutively as they attended the clinics, representing a consecutive form of convenience sampling that reduced staff discretion. This approach was particularly suitable for the present study, as obtaining a complete sampling frame in clinical settings proved challenging. Moreover, this method facilitated the timely and efficient collection of data.

### 2.4. Data Collection Steps

Upon their first or follow-up visit, the participants were recruited from the selected clinic. The healthcare providers and the receptionists’ help was sought to identify the eligible participants and distribute the questionnaire. Standardized training was provided to those who assisted with data collection in the clinics. The proposal was submitted to the College of Medicine Scientific IRB of Jouf University (HAP-13-S-001). Once ethical clearance and other necessary approvals were received (on 25 January 2025) from the relevant authorities, the data collection process commenced. The authors adhered to the principles of the Declaration of Helsinki throughout the study. To maintain confidentiality, no identifying information was collected, and participants completed the electronic questionnaire privately. All responses were stored anonymously in a password-protected database with no link to clinic staff, visit records, or personal identifiers.

Firstly, the participants were briefed about the study, its objectives, and their role as outlined in the informed consent form. Those who agreed to participate through informed consent were then invited to complete the Google form, which contained a pretested and validated questionnaire [28]. To ensure applicability in the local setting, a pilot study was conducted among 32 participants prior to data collection. The tool demonstrated acceptable internal consistency in our pilot sample (Cronbach’s α = 0.791), which was comparable to the reliability reported in the original validation study (α = 0.758) [28]. Both the original validation and our pilot testing were conducted using the Arabic version of the questionnaire. Following this verification, the tool was administered in the main study. The first part inquired about participants’ sociodemographic characteristics, and the next section asked about cosmetic procedure history, including past and planned procedures, their types, and frequencies. The third section inquired about the daily time spent on social media, the frequently used platforms, exposure to content related to cosmetic surgery, the influence of social media on self-perception, and the impact of beauty filters or trends. The final section of the questionnaire was used to screen for BDD, utilizing a validated tool. Each item probes a different aspect of appearance concern, distress, functional interference, or time spent on preoccupations. (e.g., “Are you very concerned about the appearance of some part(s) of your body?” “Has your defect(s) significantly interfered with your social life?”). The BDD screening tool uses a structured scoring system in which each item contributes to a total score ranging from 0 to 4. A total (or item) score of 4 is considered a positive screening for BDD, whereas scores from 0 to 3 are categorized as negative for BDD (i.e., no BDD).

### 2.5. Data Analysis

The data were exported from the Google Form to an Excel sheet. The Statistical Package for the Social Sciences (SPSS), version 21.0 (IBM Corp., Armonk, NY, USA), was used for coding, recoding, and further analysis. Descriptive data on demographics, BDD evaluation, and other relevant information were presented as counts and frequencies for the qualitative variables. The collected data were subjected to a test of normality, and multivariate logistic regression analysis was applied to identify factors associated with positive screening for BDD among women seeking cosmetic surgery. All covariates were entered simultaneously into the multivariable logistic regression model using the enter method. A *p*-value less than 0.05 was considered statistically significant.

## 3. Results

During the data collection time, the researchers contacted 293 eligible participants and 250 of them responded (Response rate: 85.3%) Of the 250 female participants included in the study, nearly half were aged between 26 and 40 years (48.0%), the majority of participants were single (52.4%), and nearly two-thirds (67.6%) had attained university-level or higher education. In terms of occupation, most were unemployed (27.2%), whereas 15.2% worked in government and 21.2% in the private sector. More than half of the respondents (53.2%) reported a monthly income above 10,000 SAR, and the majority were non-smokers (86.8%) (Table 1).

Regarding participants’ previous cosmetic surgical history and current surgical intentions, about 30% of the participants are revisiting the clinic, one-fourth (25.2%) had undergone cosmetic surgery earlier, and more than 36% intend to undergo surgery to correct the defects (Table 2).

Figure 1 depicts the proportion of cosmetic procedures among participants who underwent at least one cosmetic surgery. The most commonly performed surgery (34.9%), followed by liposuction (28.6%), ptosis repair (25.4%), and dacryocystorhinostomy (22.2%).

Regarding social media exposure, about 44% used more than 3 h per day. The most commonly used social media platform was Instagram (72%), about 68% exposed to cosmetic procedure content, and 28.8% mentioned that social media influenced the decision to undergo a cosmetic procedure (Table 3).

We found that more than three-fourths (76.8%) of the study participants were concerned about some aspects of their appearance, and 69.2% admitted that these concerns frequently occupied their thoughts. Furthermore, about 45% of them had received comments from others about their appearance, 42.0% experienced emotional distress due to these concerns, and 16.0% spent more than three hours per day thinking about their appearance (Table 4).

Among the 250 participants, 72 (28.8%) screened positive for BDD (Figure 2). Table 5 presents the predictors associated with BDD among females who sought cosmetic surgery. The multivariate analysis found that BDD was significantly lower among the participants aged 26 to 40 years (ref: less than 25 years, adjusted odds ratio [AOR] = 0.341, 95% confidence interval [CI] = 0.212–0.517, *p* = 0.008) and those who were more than 40 years old (ref: less than 25 years, AOR = 0.632, 95% CI = 0.481–0.019). The significantly higher levels of BDD were observed among the participants with a qualification of university and above (ref: up to high school, AOR = 1.291, 95% CI = 1.016–1.667, *p* = 0.038), who considered cosmetic surgery during their current visit (ref: no, AOR = 3.123, 95% CI = 1.671–4.982, *p* = 0.001), more than 3 h use of social media (ref: less than 1 h, AOR = 4.368, 95% CI = 3.570–5.134, *p* = 0.007), and with a prior history of cosmetic surgery (ref: no, AOR = 3.902, 95% CI = 1.719–6.284, *p* = 0.001).

## 4. Discussion

Body image dissatisfaction and cosmetic procedure demand are rising globally, particularly among young women influenced by social media exposure, highlighting the need to identify individuals at risk of BDD [2,29]. The current study determined the prevalence and predictors of screening positive for BDD among Saudi females seeking cosmetic procedures in the social media era. This study found that 28.8% of participants screened positive for BDD. Younger age, higher education, previous cosmetic procedures, current surgical intention, and greater social media use were associated with higher odds of positive screening. These findings highlight key demographic and behavioral factors linked to appearance-related concerns among women seeking cosmetic procedures.

Among the present study participants, approximately 30% were revisiting the clinic, 25.2% had previously undergone cosmetic surgery, and over one-third (36%) intended to undergo a procedure during their current visit. Rhinoplasty was identified as the most frequently performed surgery among those with prior experience. Similarly to the present study, Rammal A et al. reported that a high proportion of participants who screened positive for BDD desired for rhinoplasty [30]. Similarly, an Iranian study also reported that rhinoplasty was the most common type of cosmetic surgery (even though a much lower proportion) [31]. These findings align with the global studies, as with a high prevalence of positive screening for BDD among rhinoplasty candidates, high revisit rates for nasal revisions are regularly reported in Western cohorts, highlighting the psychological as well as aesthetic aspects of repeat procedures [32].

However, in some studies, rhinoplasty and nasal aesthetic corrections were not the most common procedures [28,33]. The variation across studies in the revisit proportion and types of procedures could be due to socio-cultural differences, accessibility of cosmetic services, and heightened appearance-related anxiety. Differences across regions may also stem from sample type (clinic-based vs. population-based, all adults vs. only females), screening methods, and post-pandemic social media influence. Furthermore, the intention to undergo a cosmetic procedure among the study participants was significantly higher than that reported by Abu Taleb R et al. [34]. From a policy perspectives, these results highlight the importance of a uniform preoperative psychological screening that is appropriate to the local setting and referral channels of repeat seekers, especially those who seek revision rhinoplasty. The present study found that greater social media use was associated with a higher likelihood of screening positive for BDD among women seeking cosmetic procedures. Studies in Saudi Arabia report very heavy social media use among cosmetic patients. For example, a study of surveyed Saudis found that about 40% spent ≥4 h per day on social media [35]. Snapchat, in particular, was found to be the most influential platform for seeking cosmetic procedures [36]. Likewise, a recent Saudi survey found a high proportion of respondents agreed that aesthetic procedures are popular among social media influencers, and 37.9% said “before-and-after” images affected their own desire for surgery [37]. Instagram use is also pervasive in Saudi cosmetic seekers (72% in our study used Instagram), reflecting a blend of both international and local platform trends [38]. The extensive use of social media, particularly long-term and daily use, by individuals seeking cosmetic surgery, highlights the necessity of organized policy changes (both public health and professional policies). At the national level, health and media regulatory bodies in Saudi Arabia and the GCC may collaborate to establish ethical guidelines for advertising cosmetic procedures on platforms such as Instagram, Snapchat, and TikTok. It is also important to note that the relationship between social media exposure and BDD symptoms may be bidirectional. While heavy social media use may heighten appearance-related concerns, individuals with pre-existing BDD tendencies may also seek out appearance-focused content more frequently.

The present study found that approximately 29% had a positive BDD screen. This finding demonstrates that the positive screening for BDDs among the cosmetic surgery-seeking patients is on an increasing trend as compared to the previous study [28,39]. One possible explanation may relate to the widespread increase in social media exposure in the post-pandemic era, although causality cannot be inferred from this cross-sectional design. Varying findings across regions highlight the need for context-appropriate interventions informed by international evidence [33,40,41].

In the present study, we found that younger participants and those who graduated from higher levels had significantly higher levels of positive screening for BDD than others. However, the age profile of Saudi BDD cases is mixed. Al Shuhayb et al. found that the majority of BDD-positive patients in a Riyadh facial surgery cohort were young adults [16]. In contrast, a Riyadh study by Munaret et al. noted that BDD-positive individuals were actually older on average (mean 34.1 vs. 30.3 years, *p* = 0.049) [42], and another study found no significant age effect [43]. Although contrasting findings were noted across the studies, age was noted as an important predictor for positive screening for BDD in some studies from other countries [7,41,44]. This suggests that BDD is skewed toward younger women. In contrast to the present study, findings on education from other studies are also inconsistent. In the facial surgery sample, the majority of BDD-positive screening patients held a college degree, suggesting higher education was common in BDD-positive screening participants. By contrast, Munaret et al. found no statistically significant difference by education level between participants with and without BDD [42]. Similarly, another study reported no education effects on BDD [43].

The present study found that those who considered cosmetic surgery during their current visit and had a prior history of cosmetic surgery had significantly higher levels of positive BDD screening. These critical associations highlights the importance of BDD screening among individuals who have had a prior visit or have consulted a specialist for cosmetic surgery evaluation, underscoring the importance of implementing uniform screening tools and coordinated multidisciplinary management to effectively address the multifaceted nature of BDD [7,19,41,45]. Similarly, those who had positive screening for BDD had a higher possibility of accepting cosmetic surgeries and had a significantly higher intention towards cosmetic surgeries than those without BDD [19,29,46]. The present study observed a finding and highlights the need for stricter ethical guidelines and pre-procedural mental-health evaluations in cosmetic settings to minimize unnecessary or potentially harmful surgeries among individuals with BDD.

Similarly to our study, one of the most common associated factors for positive screening for BDD among patients seeking cosmetic surgery is the influence of social media [29,47]. In addition to earlier suggested policy implications related to social media, there should be strict requirements that aesthetic facilities disclose when before-and-after pictures are enhanced digitally or are promotional to reduce the formation of false expectations. Integrating digital-health literacy modules into the courses of public health and school education would empower people to think critically about aesthetic information online and avoid making immediate decisions based on visual comparison. Social-media professionalism codes issued by professional organizations (e.g., the Saudi Society of Plastic Surgery) should obligate surgeons not to depict procedures in an attractive manner, but to provide balanced risk-benefit information in their posts. Lastly, collaboration between universities, ministries, and digital-platform analytics teams could help monitor trends and guide evidence-based regulation of cosmetic procedure marketing.

## 5. Strengths and Limitations

The current research adds recent evidence of BDD in Saudi women seeking cosmetic surgeries in the social media era, a region where the data is scarce. One of the study’s strengths is its use of a validated screening instrument with an acceptable level of reliability for classifying individuals with probable BDD, and the application of multivariate logistic regression. Moreover, the study incorporated the up-to-date contextual variables, including social media exposure and cosmetic surgery intention, which offered significant information regarding clinical and general healthcare policy.

However, the findings should be interpreted with caution due to certain limitations. This study is limited by its cross-sectional design, which does not establish causality between predictors and the positive screening for BDD. Additionally, the use of convenience sampling and restriction to clinics in the Eastern Province may limit the generalizability of the prevalence estimate. Women attending these clinics may differ from those in other regions in their cosmetic concerns or social media exposure, which could lead to either overestimation or underestimation of the true prevalence. The screening instrument refers to likely cases of BDD rather than confirmed clinical diagnoses, and the use of self-report information can have caused recall and/or social-desirability bias. A definitive diagnosis would require a clinician-administered gold-standard interview, such as the SCID or the BDD Diagnostic Module. Also, there was no measurement of psychological variables like depression, anxiety, or self-esteem, which restricted the measurement of psychosocial correlates.

## 6. Conclusions

The current research found that nearly one-third of Saudi females who sought cosmetic surgeries in the Eastern Province were positively screened with BDD. We also identified several factors associated with a positive screening for BDD. The present findings suggest the importance of integrating standard BDD screening measures into preoperative evaluations in cosmetic and dermatologic facilities to identify individuals at risk prior to elective surgeries. Development of multidisciplinary teamwork among dermatologists, plastic surgeons, and mental health specialists may help ensure ethical practice, improve patient safety, and reduce the number of repeated or unnecessary interventions. Multicentric and longitudinal research activities are recommended to confirm the observed association with the predictors and inform evidence-based interventions of BDD among females seeking cosmetic surgeries. These results should be interpreted within the context of a regional, convenience-based sample, and further research in other regions of Saudi Arabia is encouraged to enable broader generalization.

## Figures and Tables

**Figure 1 healthcare-13-03232-f001:**
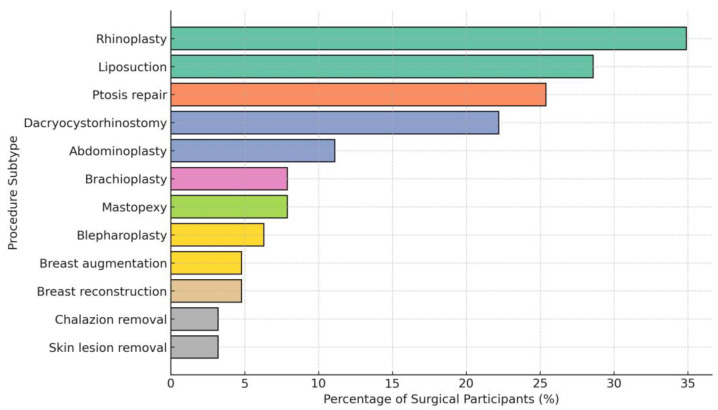
Types of cosmetic procedures performed among participants who had undergone at least one procedure (n = 63).

**Figure 2 healthcare-13-03232-f002:**
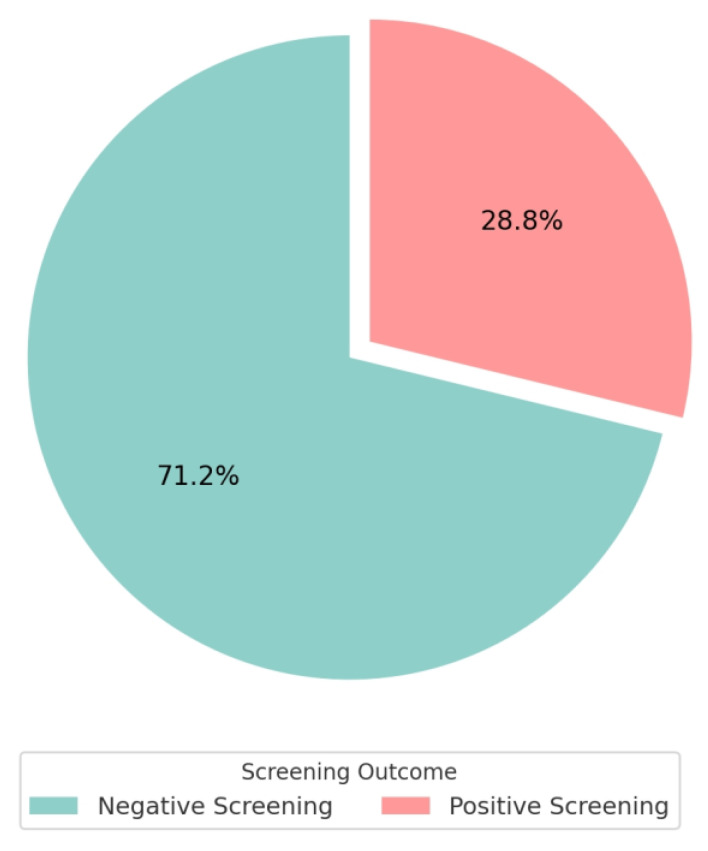
BDD screening outcomes among participants (n = 250).

**Table 1 healthcare-13-03232-t001:** Sociodemographic characteristics of the study participants (n = 250).

Characteristics	Frequency	Percentage
Age		
Less than 25 years	107	42.8
26 to 40 years	120	48.0
More than 40 Years	23	9.2
Work Status		
Unemployed	68	27.2
Government employee	69	27.6
Private sector employee	48	19.2
Self-employed	65	26.0
Marital status		
Single	131	52.4
Married	82	32.4
Divorce/widowed	38	15.2
Education level		
Up to high school	81	32.4
University and above	169	67.6
Monthly income		
Less than 5000 SAR	48	14.4
5000 to 10,000 SAR	105	32.4
More than 10,000 SAR	97	53.2
Smoking status		
No	217	86.8
Yes	33	13.2
Chronic disease		
No	164	65.6
Yes	86	34.4

**Table 2 healthcare-13-03232-t002:** Participants’ cosmetic surgery history and current surgical intentions (n = 250).

Variable	Frequency	%
Have you ever visited a consultant or clinic for plastic surgery before?		
No	174	69.6
Yes	76	30.4
Have you undergone any cosmetic surgery?		
No	187	74.8
Yes	63	25.2
Are you considering undergoing a cosmetic procedure to correct a defect during this visit?		
No	160	64.0
Yes	90	36.0

**Table 3 healthcare-13-03232-t003:** Social media use among the study participants (n = 250).

Variable	Frequency	%
Hours per day spent on social media		
Less than 1 h	45	18.0
1–3 h	95	38.0
More than 3 h	110	44.0
Social media platforms *		
Instagram	180	72.0
Snapchat	150	60.0
TikTok	138	55.2
Twitter (X)	85	34.0
Facebook	55	22.0
Other	20	8.0
Exposed to cosmetic-procedure content on social media		
Yes	170	68.0
No	80	32.0
Social media influenced the decision to undergo a cosmetic procedure		
Yes	72	28.8
No	123	49.2
Not sure	55	22.0
Beauty filters or social media trends cause dissatisfaction		
Never	35	14.0
Rarely	58	23.2
Sometimes	85	34.0
Frequently	50	20.0
Always	22	8.8

* Percentages are based on the total sample (n = 250). Multiple responses were allowed for platform selection; therefore, totals exceed 100%.

**Table 4 healthcare-13-03232-t004:** Participants’ responses to the BDD screening questionnaire items (n = 250).

Variable	Frequency	%
Are you highly concerned about the appearance of any part(s) of your body that you believe look unattractive? (BDD-Q1)		
No	58	23.2
Yes	192	76.8
Do these appearance-related concerns often occupy your thoughts to the extent that you wish you could think about them less? (BDD-Q2)		
No	77	30.8
Yes	173	69.2
Have other people ever commented on or pointed out the area(s) of your appearance that concern you? (BDD-Q3)		
No	138	55.2
Yes	112	44.8
Have these perceived defects caused you considerable distress, discomfort, or emotional pain? (BDD-Q4)		
No	145	58.0
Yes	105	42.0
Have these concerns noticeably interfered with your social relationships or social activities? (BDD-Q5)		
No	194	77.6
Yes	56	22.4
Have your appearance-related worries affected your schoolwork, job performance, or ability to fulfill your daily responsibilities? (BDD-Q6)		
No	186	74.4
Yes	64	25.6
On average, how much time per day do you spend thinking about the part(s) of your appearance that concern you? (BDD-Q7)		
Less than 1 h a day	146	58.4
1 to 3 h a day	64	25.6
More than 3 h a day	40	16.0

**Table 5 healthcare-13-03232-t005:** Predictors of positive screening of BDD among study participants (n = 250).

Variables	Total n = 250	Multivariable Analysis				
		No n = 178	Yes n = 72	Adjusted Odds Ratio	95% CI	*p*-Value
Age						
Less than 25 years	107	65	42	Ref		
26 to 40 years	120	96	24	0.341	0.212–0.517	0.008
More than 40 years	23	17	6	0.632	0.481–0.831	0.019
Work Status						
Unemployed	68	49	19	Ref		
Government employee	69	48	21	1.252	0.371–4.220	0.717
Private sector employee	48	34	14	1.950	0.672–5.663	0.219
Self-employed	65	47	18	2.152	0.639–7.246	0.216
Marital status						
Single	131	96	35	Ref		
Married	81	53	28	2.717	0.493–14.986	0.251
Divorce/widowed	38	29	9	1.593	0.353–7.186	0.545
Education level						
Up to high school	81	63	18	Ref		
University and above	169	115	54	1.291	1.016–1.667	0.038
Monthly income						
Less than 5000 SAR	48	35	13	Ref		
5000 to 10,000 SAR	105	72	33	0.666	0.253–1.755	0.411
More than 10,000 SAR	97	71	26	0.836	0.373–1.870	0.662
Smoking status						
No	217	157	60	Ref		
Yes	33	21	12	0.598	0.264–1.356	0.218
Chronic diseases						
No	209	147	62	Ref		
Yes	41	31	10	1.637	0.723–3.705	0.237
Considering a cosmetic procedure during this visit						
No	160	140	20	Ref	1.671–4.982	0.001
Yes	90	38	52	3.123		
Social media use duration						
Less than 1 h	45	40	5	Ref		
1 to 3 h	95	75	20	0.872	0.731–3.04	0.281
More than 3 h	110	63	47	4.368	3.570–5.134	0.007
Prior cosmetic surgery history						
No	187	142	45	Ref		
Yes	63	36	27	3.902	1.719–6.284	0.001

## Data Availability

The raw data supporting the conclusions of this article will be made available by the authors on request.

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
