# Peer review of "Prevalence and Predictors of Positive Screening of Body Dysmorphic Disorder in Eastern Saudi Women Seeking Cosmetic Procedures: Implications for Clinical Practice in the Social Media Era"

_healthcare, 2025, doi:10.3390/healthcare13243232_

Round 1

Reviewer 1 Report

Comments and Suggestions for Authors

Welcome and relevant manuscript addressing gaps in the literature on BDD and cosmetic surgery, which focuses on social media dynamics (a particularly valuable variable).

Methods Comments:
- Convenience sampling and regional restriction are acknowledged in the limitations but it would be helpful to add how this may influence prevalence estimates (particularly because this estimate was used to estimate power).
- Clarify whether participants were recruited consecutively or through staff discretion and if selected, add to the discussion around selection bias.
- How was confidentiality maintained in small settings?
- Is there a citation for the BDD questionnaire? Was internal consistency reverified during the study?

Results:
- Table 5, the alignment of the OR for social media variables makes it difficult to interpret the results.
- we covariates entered simultaneously in the model?

Discussion
- Difficult to say that a high rate of positive screening is equivalent to a clinical diagnosis of BDD, this is suggested several times in the discussion. This language needs to be cleaned up so as not to overstate the prevalence of clinical diagnosis.

-in the same token, language implying causality should be tempered given a cross-sectional design. 

Language
- Could be edited for redundant or awkward language (for example clarity and conciseness among figure legends and table titles)

Author Response

Authors’ reply/modifications according to the reviewer 1 comments/suggestions

General:

The authors would like to thank the reviewer for the precious time spent reviewing the paper and his excellent suggestions for improving it. Efforts have been made to modify the paper as per the reviewer’s suggestions and recommendations. The authors will be happy to hear a positive reply. All the points included according to the reviewer’s comments can be seen in track changes.

Specific response to the reviewer’s suggestions:

Kindly find the attached response to each question one by one:

Point 1: Welcome and relevant manuscript addressing gaps in the literature on BDD and cosmetic surgery, which focuses on social media dynamics (a particularly valuable variable).

Response 1: Thanks for the comment. The authors are pleased to hear the positive comments of the reviewer on the significance of the topic.

Point 2: Methods Comments:

- Convenience sampling and regional restriction are acknowledged in the limitations but it would be helpful to add how this may influence prevalence estimates (particularly because this estimate was used to estimate power).

Response 2: Thanks for the comment. According to the reviewer’s suggestions, the authors have now expanded the limitations section to clarify how the use of convenience sampling and restriction to a single region may influence prevalence estimates. Specifically, we added text explaining that this sampling approach may overestimate or underestimate BDD prevalence because the characteristics and motivations of women attending cosmetic clinics in one region may not fully represent the broader Saudi female population.

Point 3: Methods Comments:

- Clarify whether participants were recruited consecutively or through staff discretion and if selected, add to the discussion around selection bias.

Response 3: Thanks for the comment. According to the reviewer’s comments, we clarified in the revised manuscript. Participants were recruited consecutively as they attended the selected clinics, which represents a consecutive form of convenience sampling. This approach minimized staff discretion and ensured that all eligible individuals presenting during the data collection period were invited to participate.

Point 4: Methods Comments:

- How was confidentiality maintained in small settings?

Response 4: Thanks for the comment. According to the reviewer’s comment, we have clarified it in the revised manuscript. “To maintain confidentiality, no identifying information was collected, and participants completed the electronic questionnaire privately. All responses were stored anonymously in a password-protected database with no link to clinic staff, visit records, or personal identifiers.”

Point 5: Methods Comments:

- Is there a citation for the BDD questionnaire? Was internal consistency reverified during the study?

Response 5: Thanks for the comment. According to the reviewer’s comments, we included the relevant citations in the revised manuscript. Furthermore, we conducted a pilot study before proceeding to the main study to reverify it. The authors included details in the revised manuscript.

Point 6: Results:

- Table 5, the alignment of the OR for social media variables makes it difficult to interpret the results.

Response 6: Thanks for the comment. According to the reviewer’s comments, the alignment is formatted/corrected in the revised manuscript.

Point 7: Results:

- we covariates entered simultaneously in the model?

Response 7: Thanks for the comment. The authors used the enter method, in which all covariates are entered simultaneously. According to the reviewer’s comments, we clarified it in the revised manuscript method’s section.

Point 8: Discussion

- Difficult to say that a high rate of positive screening is equivalent to a clinical diagnosis of BDD, this is suggested several times in the discussion. This language needs to be cleaned up so as not to overstate the prevalence of clinical diagnosis. -in the same token, language implying causality should be tempered given a cross-sectional design.

Response 8: Thanks for the comment. We agree with the reviewer that screening-positive cases cannot be interpreted as clinical diagnoses of BDD. According to the reviewer’s suggestions, the Discussion section has been revised to ensure that all statements refer specifically to “positive screening for BDD” rather than “BDD” as a formal diagnosis. Furthermore, the authors have also softened any language that may imply causality, replacing it with terms that reflect associations only (considering the present study design).

Point 9:

Language

- Could be edited for redundant or awkward language (for example clarity and conciseness among figure legends and table titles).

Response 9: Thanks for the comment. According to the reviewer’s comments, the manuscript has undergone careful language editing by a native English speaker to improve clarity, remove redundancy, and enhance readability. Furthermore, we corrected sentences, grammar, and other language issues using Grammarly Premium (paid version). Several long or repetitive sentences in the Introduction and Discussion were refined, and table titles and figure legends were made more concise.

 The authors thank the reviewer once again for the positive and constructive comments.

Reviewer 2 Report

Comments and Suggestions for Authors

The manuscript addresses an important and timely topic, and the study is overall well-structured. The introduction provides extensive background, but it is somewhat repetitive in several sections, especially regarding social media influences and cultural beauty norms. A more concise and focused presentation of the knowledge gap would strengthen the narrative and better highlight the contribution of this study.

The methodological description is generally adequate, yet the section would benefit from greater clarity and conciseness. The explanation of the BDD screening tool is overly detailed and could be streamlined, particularly the scoring system. It would also be helpful to clarify whether the questionnaire was validated in Arabic and whether any pilot testing was conducted locally. In addition, the use of convenience sampling should be more explicitly justified, as this choice directly impacts generalizability.

The results are clearly organized, with tables and figures that are easy to interpret. However, some percentages in Table 3 appear inconsistent and may require verification or clarification regarding missing or excluded responses. Minor formatting issues in the logistic regression table could also be corrected to improve readability.

The discussion is comprehensive and successfully contextualizes the findings within the existing literature, but the section would benefit from a more concise synthesis of the key results before transitioning into comparisons with other studies. Some paragraphs are lengthy and could be refined for better flow. The policy considerations are thoughtful and relevant, although presenting them more succinctly would make this section more impactful. It may also be useful to comment more explicitly on the possible bidirectional relationship between social media exposure and BDD symptoms, as this nuance adds depth to the interpretation.

The conclusions are appropriate and aligned with the data, yet they should be phrased with slightly more caution to avoid broad generalizations about national implications that go beyond the study’s sampling design. 

Comments on the Quality of English Language

The English language is generally understandable; however, several sections—particularly within the Introduction and Discussion—contain long or repetitive sentences that affect clarity and flow. A moderate level of editing would enhance the overall quality of the manuscript.

Author Response

Authors’ reply/modifications according to the reviewer 2 comments/suggestions

General:

The authors would like to thank the reviewer for the precious time spent reviewing the paper and his excellent suggestions for improving it. Efforts have been made to modify the paper as per the reviewer’s suggestions and recommendations. The authors will be happy to hear a positive reply. All the points included according to the reviewer’s comments can be seen in track changes.

Specific response to the reviewer’s suggestions:

Kindly find the attached response to each question one by one:

Point 1:The manuscript addresses an important and timely topic, and the study is overall well-structured. The introduction provides extensive background, but it is somewhat repetitive in several sections, especially regarding social media influences and cultural beauty norms. A more concise and focused presentation of the knowledge gap would strengthen the narrative and better highlight the contribution of this study.

Response 1: Thanks for the positive comments. The authors are pleased to hear the reviewer's positive comments on the importance of the topic. According to the reviewer’s suggestions, the repetitive contents have been deleted in the revised manuscript.

Point 2: The methodological description is generally adequate, yet the section would benefit from greater clarity and conciseness. The explanation of the BDD screening tool is overly detailed and could be streamlined, particularly the scoring system. It would also be helpful to clarify whether the questionnaire was validated in Arabic and whether any pilot testing was conducted locally. In addition, the use of convenience sampling should be more explicitly justified, as this choice directly impacts generalizability.

Response 2: Thanks for the positive comments on methods. According to the reviewer’s suggestions, the scoring system is shortened in the revised manuscript. Both the original validation and our pilot testing were conducted using the Arabic version of the questionnaire. The details are included in the revised manuscript. Justifications for convenience sampling and limitations of the choice are expanded in the revised manuscript.

Point 3: The results are clearly organized, with tables and figures that are easy to interpret. However, some percentages in Table 3 appear inconsistent and may require verification or clarification regarding missing or excluded responses. Minor formatting issues in the logistic regression table could also be corrected to improve readability.

Response 3: Thanks for the positive comments on the results. According to the reviewer’s comments, we carefully rechecked all frequencies and percentages in Table 3 and confirmed that they are accurate. We would also like to clarify that the “social media platforms” section allowed multiple responses, as participants could select more than one platform; therefore, the percentages in this category do not sum to 100%. This clarification has now been added to the table footnote. Minor formatting adjustments were also made to enhance readability.

Point 4: The discussion is comprehensive and successfully contextualizes the findings within the existing literature, but the section would benefit from a more concise synthesis of the key results before transitioning into comparisons with other studies. Some paragraphs are lengthy and could be refined for better flow. The policy considerations are thoughtful and relevant, although presenting them more succinctly would make this section more impactful. It may also be useful to comment more explicitly on the possible bidirectional relationship between social media exposure and BDD symptoms, as this nuance adds depth to the interpretation. 

Response 4: Thanks for the positive comments on the discussion. According to the reviewer’s comment. According to the reviewer’s suggestions, the key findings (concise) are added before we proceed to the main comparison (paragraph 1 in the discussion section). We also shortened the length of paragraphs and sentences. Furthermore, the possible bidirectional relationship was also discussed in the revised manuscript.

Point 5: The conclusions are appropriate and aligned with the data, yet they should be phrased with slightly more caution to avoid broad generalizations about national implications that go beyond the study’s sampling design. 

Response 5: Thanks for the positive comments on the conclusion. According to the reviewer’s comments, we revised the conclusion.

Point 6: The English language is generally understandable; however, several sections—particularly within the Introduction and Discussion—contain long or repetitive sentences that affect clarity and flow. A moderate level of editing would enhance the overall quality of the manuscript. 

Response 6: Thanks for the comment. According to the reviewer’s comments, the Introduction and Discussion have been carefully reviewed, and several long or repetitive sentences have been shortened or restructured to improve clarity and flow. Additionally, the manuscript has undergone careful language editing by a native English speaker to improve clarity, remove redundancy, and enhance readability. Furthermore, we corrected sentences, grammar, and other language issues using Grammarly Premium (paid version).

The authors thank the reviewer once again for the positive and constructive comments.

Reviewer 3 Report

Comments and Suggestions for Authors

The authors have presented their interesting work on prevalence and predictors of body dysmorphic disorder screening among Eastern Saudi women. It is a cross-sectional study and the methodology is well described.

Body Dysmorphic Disorder (BDD)is a condition characterized by a distressing preoccupation with perceived defects in appearance that are slight or non-existent to others. It has been found to disproportionately affect individuals seeking cosmetic interventions. Prevalence of BDD among population cohort seeking cosmetic surgery varies widely in the literature, which could be because of the reluctance or inability of patients to respond honestly to the screening questions. Even though the authors have used a pre-validated tool to screen BDD with high internal consistency, self-reported screening tools for BDD have these inherent flaw.   The "gold standard" for diagnosing BDD is the Structured Clinical Interview for DSM-5 (SCID) or the specific BDD Diagnostic Module administered by a trained clinician. (1)

The definition of cosmetic surgery needs to be placed clearly in order to conduct this study. The examples given in Figure 1 on the types of procedures include procedures like dacryocystorhinostomy, ptosis repairs, Chalazion removal as well as skin lesion removal. These surgeries have more of a reconstructive component and are not pure cosmetic surgeries. Among rhinoplasty cases, it is important to understand how many were pure cosmetic rhinoplasty and not due to some secondary conditions like post-traumatic deformities or binder syndrome.  I would like the authors to respond to these comments and add their insights to the revised manuscript. 

Although the methodology is not robust, particularly the use of self-reported screening tool by the patients and the selection of cases which are considered as cosmetic surgery by the authors, nevertheless, the study results throw some insight on the prevalence of BDD among this cohort. The manuscript is generally written well. The authors are strongly encouraged to carry this research forward with a stronger research design. probably a prospective study with a wider cohort of population. 

 Reference:
1. Pereira IN, Chattopadhyay R,Fitzpatrick S, Nguyen S, Hassan H. Evidence-based review:Screening body dysmorphic disorder in aesthetic clinicalsettings. J Cosmet Dermatol. 2023;22:1951-1966. doi:10.1111/jocd.15685

Author Response

Authors’ reply/modifications according to the reviewer 3 comments/suggestions

General:

The authors would like to thank the reviewer for the precious time spent reviewing the paper and his excellent suggestions for improving it. Efforts have been made to modify the paper as per the reviewer’s suggestions and recommendations. The authors will be happy to hear a positive reply. All the points included according to the reviewer’s comments can be seen in track changes.

Specific response to the reviewer’s suggestions:

Kindly find the attached response to each question one by one:

Point 1: The authors have presented their interesting work on prevalence and predictors of body dysmorphic disorder screening among Eastern Saudi women. It is a cross-sectional study and the methodology is well described. 

Response 1: Thanks for the comments. The authors are pleased to hear the positive comments from the reviewer regarding the significance of the work and methods.

Point 2:  Body Dysmorphic Disorder (BDD)is a condition characterized by a distressing preoccupation with perceived defects in appearance that are slight or non-existent to others. It has been found to disproportionately affect individuals seeking cosmetic interventions. Prevalence of BDD among population cohort seeking cosmetic surgery varies widely in the literature, which could be because of the reluctance or inability of patients to respond honestly to the screening questions. Even though the authors have used a pre-validated tool to screen BDD with high internal consistency, self-reported screening tools for BDD have these inherent flaw.   The "gold standard" for diagnosing BDD is the Structured Clinical Interview for DSM-5 (SCID) or the specific BDD Diagnostic Module administered by a trained clinician. (1).

Response 2: Thanks for the comment. We agree with the reviewer, and we have enhanced the limitations as per the reviewer’s suggestions. Furthermore, the authors critically revised not to overstate the findings and made it (in revised manuscript) “positive screening for BDD” rather than a clinical diagnosis throughout the manuscript.

Point 3: The definition of cosmetic surgery needs to be placed clearly in order to conduct this study. The examples given in Figure 1 on the types of procedures include procedures like dacryocystorhinostomy, ptosis repairs, Chalazion removal as well as skin lesion removal. These surgeries have more of a reconstructive component and are not pure cosmetic surgeries. Among rhinoplasty cases, it is important to understand how many were pure cosmetic rhinoplasty and not due to some secondary conditions like post-traumatic deformities or binder syndrome.  I would like the authors to respond to these comments and add their insights to the revised manuscript.

Response 3: Thanks for the comment. We agree that a clear operational definition of cosmetic procedures strengthens the methodological clarity of the study. According to the reviewer’s suggestion, we have added a definition in the Methods section specifying that cosmetic procedures were considered elective aesthetic interventions sought primarily to enhance appearance. We excluded those seeking procedures (including rhinoplasty) primarily for functional, reconstructive, or medical reasons rather than cosmetic purposes.

Point 4: Although the methodology is not robust, particularly the use of self-reported screening tool by the patients and the selection of cases which are considered as cosmetic surgery by the authors, nevertheless, the study results throw some insight on the prevalence of BDD among this cohort. The manuscript is generally written well. The authors are strongly encouraged to carry this research forward with a stronger research design. probably a prospective study with a wider cohort of population.

Response 4: Thanks for the comment and thoughtful encouragement to strengthen this line of research. We have clarified the methodological limitations in the revised manuscript, including the use of self-reported screening tools and the definition of cosmetic procedures. We also appreciate the suggestion for more robust designs, and the authors intend to pursue future prospective studies with wider and more diverse cohorts. 

 The authors thank the reviewer again for the positive and constructive comments.